# Ecological, Social, and Other Environmental Determinants of Dengue Vector Abundance in Urban and Rural Areas of Northeastern Thailand

**DOI:** 10.3390/ijerph18115971

**Published:** 2021-06-02

**Authors:** Md. Siddikur Rahman, Tipaya Ekalaksananan, Sumaira Zafar, Petchaboon Poolphol, Oleg Shipin, Ubydul Haque, Richard Paul, Joacim Rocklöv, Chamsai Pientong, Hans J. Overgaard

**Affiliations:** 1Department of Microbiology, Faculty of Medicine, Khon Kaen University, Khon Kaen 40002, Thailand; tipeka@kku.ac.th; 2Department of Statistics, Begum Rokeya University, Rangpur 5404, Bangladesh; 3HPV & EBV and Carcinogenesis Research Group, Khon Kaen University, Khon Kaen 40002, Thailand; 4Environmental Engineering and Management Program, Asian Institute of Technology, Pathumthani 12120, Thailand; st120302@ait.asia (S.Z.); olegshipin@gmail.com (O.S.); 5Office of Disease Prevention and Control 10, Ubon Ratchathani 35000, Thailand; siapoolphol@gmail.com; 6Department of Biostatistics and Epidemiology, University of North Texas Health Science Center, North Texas, Fort Worth, TX 76107, USA; mdubydul.haque@unthsc.edu; 7Unité de la Génétique Fonctionnelle des Maladies Infectieuses, Institut Pasteur, CNRS UMR 2000, 75015 Paris, France; rpaul@pasteur.fr; 8Department of Public Health and Clinical Medicine, Umeå University, 90187 Umeå, Sweden; joacim.rocklov@umu.se; 9Faculty of Science and Technology, Norwegian University of Life Sciences, P.O. Box 5003, 1430 Ås, Norway

**Keywords:** *Aedes aegypti*, vector control, climate change, dengue, knowledge, attitudes, and practices (KAP), entomological indices

## Abstract

*Aedes aegypti* is the main vector of dengue globally. The variables that influence the abundance of dengue vectors are numerous and complex. This has generated a need to focus on areas at risk of disease transmission, the spatial-temporal distribution of vectors, and the factors that modulate vector abundance. To help guide and improve vector-control efforts, this study identified the ecological, social, and other environmental risk factors that affect the abundance of adult female and immature *Ae. aegypti* in households in urban and rural areas of northeastern Thailand. A one-year entomological study was conducted in four villages of northeastern Thailand between January and December 2019. Socio-demographic; self-reported prior dengue infections; housing conditions; durable asset ownership; water management; characteristics of water containers; knowledge, attitudes, and practices (KAP) regarding climate change and dengue; and climate data were collected. Household crowding index (HCI), premise condition index (PCI), socio-economic status (SES), and entomological indices (HI, CI, BI, and PI) were calculated. Negative binomial generalized linear models (GLMs) were fitted to identify the risk factors associated with the abundance of adult females and immature *Ae. aegypti*. Urban sites had higher entomological indices and numbers of adult *Ae. aegypti* mosquitoes than rural sites. Overall, participants’ KAP about climate change and dengue were low in both settings. The fitted GLM showed that a higher abundance of adult female *Ae. aegypti* was significantly (*p* < 0.05) associated with many factors, such as a low education level of household respondents, crowded households, poor premise conditions, surrounding house density, bathrooms located indoors, unscreened windows, high numbers of wet containers, a lack of adult control, prior dengue infections, poor climate change adaptation, dengue, and vector-related practices. Many of the above were also significantly associated with a high abundance of immature mosquito stages. The GLM model also showed that maximum and mean temperature with four-and one-to-two weeks of lag were significant predictors (*p* < 0.05) of the abundance of adult and immature mosquitoes, respectively, in northeastern Thailand. The low KAP regarding climate change and dengue highlights the engagement needs for vector-borne disease prevention in this region. The identified risk factors are important for the critical first step toward developing routine *Aedes* surveillance and reliable early warning systems for effective dengue and other mosquito-borne disease prevention and control strategies at the household and community levels in this region and similar settings elsewhere.

## 1. Introduction

Dengue is an emerging and re-emerging mosquito-borne viral disease of humans [1] that is caused by dengue viruses (DENV) from the *Flavivirus* genus with four serotypes: DENV 1–4. DENV is transmitted by bites of infected *Aedes* mosquitoes, specifically *Aedes* (*Ae.*) *aegypti* Linnaeus and *Ae. albopictus* Skuse, which are known as major and secondary dengue vectors and are also vectors of chikungunya, yellow fever, and Zika viruses [2,3]. The incidence of dengue has dramatically spread and increased globally in the past 40 years. Approximately half of the world’s population is at risk of contracting the disease, with an estimated 390 million infections occurring annually in 128 countries [4]. Dengue affects most of the world’s tropical and sub-tropical regions; Southeast Asia and the Western Pacific, in particular, have been seriously affected [5]. The disease is one of the main threats to public health and a leading cause of hospitalization in Thailand [6]. The first DENV infection was reported in 1949, the first outbreak was in 1958 [7,8], and several major outbreaks with high morbidity were documented in 1987, 1997, 1998, 2001, 2013, 2015, and 2019 across the country [9,10,11]. One of the largest dengue outbreaks in Thailand was in 1987 with 174,285 cases and 1008 deaths. According to the WHO, the country ranked sixth among the 30 most highly dengue-endemic countries in the world during 2004–2010 [12]. The Ministry of Public Health, Thailand, reported more than 72,000 dengue cases with a fatality rate of 0.13% in 2018. Recently, 71,292 dengue cases with 5151 deaths were reported in 2020 from the whole country, with all regions affected. The highest incidence rates (cases per 100,000 population) were found in the northeastern region [13]. All four DENV serotypes circulate in Thailand [9,14,15], and disease transmission is seasonal, with a peak in the rainy period from May to October [2]. 

The development and abundance of dengue vectors are affected by various ecological, socio-economic, and other environmental factors [14,15,16,17,18]. *Ae. aegypti* breeds in domestic water storage containers, which constitute major oviposition sites and an important risk factor for DENV transmission in Thailand [14]. Human-related factors that generate such artificial containers for larval development are important risk determinants of the distribution of dengue vectors. These include local socio-economic conditions, human habitats, and water storage practice-related behaviors. Socio-demographic factors also affect dengue vector production and transmission of DENV. For instance, the risk of DENV in Thailand was associated with people gaining at least a secondary education level and with households of more than four members [14,16]. It is well known that environmental factors influence diverse aspects of vector and virus biology by influencing mosquito population dynamics and virus circulation [19,20]. Furthermore, vector abundance and mosquito development vary seasonally because of local changes in temperature, humidity, and rainfall, all of which affect the availability of larval development sites and DENV transmission [21]. There have been several published studies on the climatic effect on vector abundance in Thailand [22,23,24,25,26,27,28], but there have been few detailed regional studies within Thailand. 

The variables that influence the vector breeding and production of *Aedes* mosquitoes are numerous and complex [29]. Therefore, an in-depth understanding of fine-scale ecological, social, and other environmental risk factors that modulate vector abundance can provide vital information to fill in the gaps of our knowledge regarding the complex dynamics of dengue vectors and associated disease risk. In the absence of effective dengue treatment options and limitations of vaccines [30,31], updated information related to dengue vector abundance and associated risk factors is also essential for designing effective vector-control programs and dengue-prevention strategies. Understanding knowledge, attitudes, and practices (KAP) related to local climate change, dengue, and vector biology and control can identify perception gaps that can help guide new community engagement tools to enhance the effectiveness of vector-control efforts and disease awareness [32,33]. Notably, data on *Aedes* abundance and its risk factors are scanty in the northeastern region of Thailand despite it being a dengue-endemic area with a high population size and a land area proportional to the whole country. This study aimed to identify the spatiotemporal distributions and abundance of *Ae. aegypti* and to determine the associated predictors across urban and rural areas in northeastern Thailand. 

## 2. Materials and Methods

### 2.1. Study Sites 

The study was conducted in four sites (two urban and two rural) in two provinces in northeastern Thailand. The selected sites were the urban Tat Khaen (16°33′18.4″ N, 104°42′01.2″ E) and rural Na Sameng (16°17′46.5″ N, 104°52′10.4″ E) in Mukdahan province and the urban Don Yung (15°17′32.9″ N, 104°49′11.8″ E) and rural Phon Duan (14°38′38.1″ N, 105°05′34.5″ E) in Ubon Ratchathani province (Figure 1). Details of the four study sites were previously described [33]. These sites were selected based on high dengue incidence during 2014–2018, feasibility, and logistics. The study area has a tropical climate with a dry season from October to April and a rainy season (dominated by the southwest monsoon) with high rainfall, high humidity, and high temperatures from May to September. DENV is mainly transmitted during the rainy season.

### 2.2. Study Design and Sample Size

The study was pursued within a larger project in Thailand and Laos (DENCLIM project; 2018–2021) that aims to evaluate the effects of climate change and variability on community vulnerability and exposure to dengue in the Southeast Asia. The total household sample size (90 households in each site) was calculated to be able to significantly detect differences in the number of dengue cases between urban and rural sites [33]. However, in this sub-study, 32 households were randomly selected for entomological, KAP, household, and climate data collection from each site, resulting in 128 households (64 urban and 64 rural). All selected households were given a unique identification number and geo-referenced using a global positioning system (GPS) device. 

### 2.3. Entomological Survey

Monthly mosquito collections were carried out indoors and outdoors in the selected households in each study site from January to December 2019. The following data were collected: the number of adult mosquitoes (all genera and species), immature mosquitoes (larvae and pupae of all genera and species), the number of total containers, the number of wet containers, the number of mosquito-positive containers, and the characteristics of all positive containers. Adult mosquitoes were collected for 10 min indoors (in main rooms of activity, e.g., living rooms and bedrooms) and 10 min outdoors (among artificial articles, cars, motorcycles, vegetation, tree holes, roof gutters, etc.) in each household using battery-powered Prokopack aspirators [34]. Mosquito larvae and pupae were collected from larger containers (water volume of approximately >30 L) by the ‘five-sweep′ procedure using a fine-mesh hand screen and by a regular larval dipper or pipette from smaller containers (water volume approximately <30 L) [35]. Collected water samples were poured through a strain into white bowls for the better visualization, counting, and collecting of specimens [36]. The number of larvae and pupae was recorded in three categories (<10, 11–100, and >100). At least 20 larvae (if available) and all pupae were collected for further processing. 

### 2.4. Mosquito Handling

Adult mosquitoes were stored in the aspirator collection cups in Styrofoam boxes and brought back to the laboratory. Mosquitoes from these samples were killed by freezing and then morphologically identified to *Ae. aegypti*, *Ae. albopictus*, *Culex* spp., or other (*Armigeres* spp./*Anopheles* spp.) using a stereomicroscope. Adult mosquitoes were sorted by sex and identified to species. Adults were stored individually in 1.5 mL micro-centrifuge tubes at −20 °C until further analysis. Immature mosquitoes were stored in separate, labeled, 100 mL Whirl-Pak plastic bags, and then they were recorded and transported to the entomology laboratory for specimen sorting and species identification. Immature *Aedes* were identified to species using morphological taxonomic keys [34,37]. 

### 2.5. Household and Demographic Survey

A cross-sectional survey was conducted in February 2019, and socio-demographic data were collected from household respondents of each selected household using a structured questionnaire. In addition to the semi-structured interviews, observations were made, and household information—including household characteristics (house condition, yard condition, shade condition, house type, and house materials), water management (water, hygiene, and sanitation facilities), and vector-control practices (adult and larvae control)—was recorded. The household survey also measured household wealth using a set of questions on durable asset ownership. This information was used to generate a measure of socio-economic status (SES). Housing density around sample households was calculated by counting houses within a 200-m buffer using Google Earth.

### 2.6. Knowledge, Attitudes and Practices Survey

Data on participants′ KAP regarding climate change and dengue were used from a broader collection of KAP surveys described in our previous study [33]. Data on climate change included knowledge about its connection to dengue, as well as local and global climate change problems, attitudes, and adaptation and mitigation practices. Data on dengue included knowledge on transmission, symptoms, and signs; most frequent bite time of mosquitoes; vector morphology and vector breeding sites; attitudes and prevention practices, such as bite prevention practices; and *Aedes* breeding prevention methods. 

### 2.7. Climate Data

Weather stations were set up at the beginning of the project in three study sites located within 1–2 km from inspected houses. Meteorological data in Tat Khaen were obtained from a nearby meteorological station of the Department of Meteorology in Mukdahan (16°32′57.6″ N, 104°42′15.4″ E 16°32′ N, 104°43′ E). Meteorological data including daily maximum, minimum, and total rainfall (mm); maximum and minimum temperatures (°C); and relative humidity (%) were collected from the weather stations during the study period and aggregated at the weekly level. 

### 2.8. Data Management and Statistical Analysis

The entomological indices calculated in this study were the house index (HI) (percentage of houses infested with larvae and/or pupae), container index (CI) (percentage of water-holding containers infested with larvae and/or pupae), Breteau index (BI) (number of positive containers per 100 houses inspected), and pupae index (PI) (number of pupae per 100 houses inspected) [36,38]. The characteristics of the most productive container types producing >60% of all pupae were also calculated. The household crowding index (HCI) was calculated using the total number of household residents (excluding newborn infants) at each monthly visit divided by the total number of rooms, excluding kitchen and bathrooms. The HCI was grouped into three categories: <1, 1–2, and >2 residents per room [39,40] (Table 1). The premise condition index (PCI) was estimated for each household and classified based on the general condition of the house, the surrounding yard area, the degree of shade, and the water management systems [41,42]. The PCI could take on a minimum value of 4 (good condition) and a maximum of 9 (bad condition) (Table 1). The SES of selected households was ranked as poor, intermediate, or wealthy. A detailed description of the method for constructing SES was provided in a previously published paper [33] (Table 1). The KAP of the 128 household respondents were assessed using the same scoring system published in our previous study [33] based on the total correct responses against the total questions (questions and summary answers shown in Appendix A).

Descriptive statistical analyses of mosquito collections, species composition, container characteristics, entomological indices, socio-demographics, KAP, and meteorological variables were conducted. Generalized linear models (GLMs) were fitted to investigate the association of socio-demographics, KAP, and household risk factors with the abundance of adult female and immature *Ae*. *aegypti* per household. The lag effect (0–4 weeks) of climatic factors (i.e., minimum, mean, and maximum temperatures, relative humidity, and rainfall) on both adult female and immature *Ae*. *aegypti* indices were also investigated using GLMs, assuming a negative binomial distribution with the logarithmic link function. The incidence risk ratio (IRR) was also calculated and adjusted using multivariable analysis. A negative binomial distribution was used since the response variables were over-dispersed count data (adult female *Ae. aegypti* per household (variance = 47.2, mean = 8.3), adult female *Ae. aegypti* per month (variance = 2261.9, mean = 88.8), immature *Ae. aegypti* per household (variance = 414.1, mean = 17.2), and immature *Ae. aegypti* per month (variance = 8368.8, mean = 1849)) [43]. Statistical analyses were performed using RStudio with the “MASS” package [44]. Figures were produced using the “carData,” “effects,” “ggplot2,” and “ggpbur” packages [45,46]. Maps of each study sites were created to visualize the number of adults and immature *Aedes* mosquitoes collected from each household over the study period. 

## 3. Results 

### 3.1. Entomological Collections and Indices 

A total of 5273 adult mosquitoes were collected. The most abundant species were *Ae. aegypti* 2658 (50.5%), followed by *Culex* spp. 1979 (37.5%) and others 561 (10.6%). *Ae. albopictus* 75(1.4%) was the least abundant. Among the 1113 female *Aedes* spp. (40.7% of the total *Aedes* collected), 1066 (95.8%) were *Ae. aegypti*. An overall monthly mean of 45.9 *Ae. aegypti* females was collected in the urban sites, and a monthly mean of 42.9 was collected in the rural sites (Table 2).

Urban sites had higher mean numbers of *Ae. aegypti* mosquitoes—all immature and pupae—than the rural sites. The highest numbers of adult *Ae. aegypti* (1374) and *Ae. albopictus* (52), including both females and males, were collected in the urban sites (Table 2); the numbers collected during the wet season (May–October) were 1516 and 51, respectively (Figure 2). During the 12 months of collections, a cumulative total of 856 out of 9399 inspected water-holding containers were found positive for immature mosquitoes. There were 2594 *Aedes* immature mosquitoes, of which 2210 (85%) were identified as *Ae. aegypti* and 384 (15%) as *Ae. albopictus* (Table 2). The highest number of immature mosquitoes (1415) was collected during the dry season (March–April), and the lowest number (1179) was collected during the wet season (Figure 2 and Figure 3). The corresponding numbers for urban and rural sites were 1518 (58.5%) and 1076 (41.5%), respectively.

Entomological indices (HI, CI, BI, and PI) were found to be higher in urban sites than rural sites (Figure 4). The overall figures of entomological indices during January–December 2019, recorded for HI, CI, BI, and PI were urban = 41.2% and rural = 34.9%, urban = 9.4% and rural = 8.8%, urban = 64.2 and rural = 47.3, and urban = 87.8 and rural = 67.0, respectively.

### 3.2. Container Characteristics and Breeding of Dengue Vectors 

Of the 856 positive containers found in both urban and rural sites during the 12 collection months of 2019, 75% of immature *Aedes* (pupae and larvae) were collected in round-shaped containers (jar, bucket, etc.), 14% were collected in square-shaped containers (cemented tank, flower vase/pots, etc.), and 11% were collected in other types of breeding sites (Table 3). Medium-sized (50–100 cm) containers were the highest infested containers in both urban and rural areas. Containers with any type of mosquito control were less infested with *Aedes* larvae (urban: 87%; rural: 81%) and pupae (urban: 90%; rural: 86%) than those without control. Containers treated with abate were the least infested containers. Uncovered and outdoor containers were the highest contributors to dengue vector breeding in both areas compared to well-covered and indoor containers. 

### 3.3. Knowledge, Attitudes, and Practices on Climate Change 

Almost all of the study respondents reported having heard about climate change (95% and 90% for urban and rural sites, respectively). More than 60% of the participants in both urban and rural sites believed that changes in climate can affect dengue fever and its vectors. However, study communities in both areas had limited knowledge and awareness about local, global, and possible future effects of climate change, as well as regarding the topics of global climate change and changing mosquito habitat suitability (Appendix A). More than 90% of the study respondents had a positive attitude to receiving updated information about the impacts of climate change and the mitigation of dengue risk (Appendix A). Only a few respondents (28% and 45% for urban and rural, respectively) took additional actions to prepare to adapt to the impact of climate change (e.g., floods, droughts, and storms) and to reduce dengue risk (Appendix A). Most of the respondents used some form of climate-resilient household practices regarding the spread of dengue due to climate change. The most common best practices among respondents included cleaning up drainage systems from waste (68.8% and 54.7% for urban and rural, respectively). Urban respondents had overall good level of KAP on climate change, with K = 28%, A = 67%, and *p* = 23%; the corresponding figures for rural respondents were K = 20%, A = 56%, and *p* = 13%. Detailed results are presented in Appendix A.

### 3.4. Knowledge, Attitudes, and Practices on Dengue 

The study respondents had limited knowledge of the transmission, symptoms, and warning signs of DENV and dengue disease (Appendix A). Less than 50% of the study respondents knew about DENV serotypes and the name of dengue vectors (*Ae. aegypti* and *Ae. albopictus)*. Regarding vector morphology, few respondents (53% and 23% in urban and rural areas, respectively) knew that *Aedes* mosquitoes have white spots on their legs. However, more than 80% of the respondents knew that these mosquitoes are daytime biters. For dengue risk mitigation, the majority showed a positive attitude of requiring improved awareness and knowledge, as well as more educational programs on symptoms and treatments of dengue, including additional training on vector-control strategies (Appendix A). In both areas, dengue-prevention practices related to *Aedes* breeding sites and steps to prevent mosquito breeding during an outbreak were not satisfactory (Appendix A). The most common best practices among respondents included preventing mosquito–human contact followed by covering and protecting skin with clothes, using window screens and bed nets, disposing water-holding containers, covering water containers, and using insecticide sprays to reduce mosquitoes. The overall percentages of the population with a good level of KAP on dengue in urban areas were K = 45%, A = 72%, and P = 30%, and the corresponding figures for rural areas were K = 40%, A = 65%, and P = 23%. Detailed results are presented in Appendix A.

Overall, KAP regarding climate change and dengue were low in urban and rural sites (KAP scores considered were good if ≥80 and poor if <80). Urban residents had higher mean KAP scores regarding climate change and dengue than rural residents (Figure 5), but there were no significant differences regarding KAP level between the two sites (Appendix A). 

### 3.5. Ecological and Social Determinants of the Abundance of Adult Female and Immature Ae. aegypti

Urban sites had a significantly higher abundance of adult female *Ae. aegypti* (IRR: 1.55; 95% confidence interval (CI): 1.21–1.98) and immature *Ae. aegypti* (IRR: 1.47; 95% CI: 1.01–2.17) than rural sites. Educational level was significantly associated with the abundance of both adult female and immature *Ae. aegypti*. The houses of respondents who had lower education levels (only primary school) were more likely to be infested with adult female (IRR: 1.39; 95% CI: 1.07–1.79) and immature (IRR: 1.49; 95% CI: 1.02–2.18) *Ae. aegypti* than the houses of respondents with primary education or higher. Poor households (lower SES) were found to be significantly associated with a higher abundance of immature *Ae. aegypti* (IRR: 2.15; 95% CI: 1.39–3.32) than wealthy households. Crowded (HCI > 3) and medium crowded (HCI > 2) households were significantly associated with a higher abundance of adult female *Ae. aegypti,* but results for immature mosquitoes were not as clear (Table 4). A higher PCI was also significantly associated with a higher abundance of both adult female (IRR: 1.97; 95% CI: 1.49–2.61) and immature (IRR: 1.54; 95% CI: 1–2.37) *Ae. aegypti*.

Houses located in clusters with a medium housing density (201–500 houses per km^2^) were more significant predictors for adult females (IRR: 1.37; 95% CI: 1.05–1.8) compared to high density settings (>1000 houses per km^2^), whereas for immature mosquitoes, this occurred in clusters with 501–1000 houses per km^2^ (IRR: 1.49; 95% CI: 1.02–2.18). Houses with unscreened windows were found to be significantly associated with a higher abundance of both mosquito stages. Houses with the bathroom located indoors, with higher numbers of water-filled containers (regardless of being mosquito-positive or not) on their premises, and without any adult control interventions were more likely than not to be infested with adult female *Ae. aegypti*. Poor climate change adaptations and dengue preventive practices (*p* < 0.05) were significantly associated with higher abundances of both mosquito stages (Table 4). 

### 3.6. Climatic Determinants of the Abundance of Adult Females and Immature Ae. aegypti 

During the study period (from January to December 2019), mean monthly temperatures ranged from 23.2 to 32.6 °C and from 22.7 to 31.0 °C in urban and rural sites, respectively. The mean relative humidity levels in the urban and rural sites were 73.5% and 76.3%, respectively. The relative humidity varied between 58.0% and 88.1% in urban sites and between 60.9% and 88.7% in rural sites. The recorded mean total rainfall values in the urban and rural sites were 102.9 and 166.5 mm, respectively. The total rainfall varied between 0 and 803.3 mm in urban sites and between 0 and 1229.4 mm in rural sites (Table 5).

There was a considerable variation in the climate results. Though most of the relationships were non-significant, the mean and maximum temperature at four weeks of lag seemed to be generally more significant for adult mosquitoes than fewer-week lags (Figure 6). Higher temperatures were also found to be related to immature numbers at more recent times near the collection event (Figure 7). Detailed results for significant associations of climate variables at different week lags with the abundance of adult and immature mosquitoes are presented in Figure 6 and Figure 7. 

## 4. Discussion

The present study investigated the spatial-temporal abundance of dengue vectors and determinants of their prevalence in the selected study areas of northeastern Thailand. *Ae. aegypti* was a more abundant dengue mosquito species than *Ae. albopictus*. *Ae. aegypti* is the main dengue vector in Thailand and is well-adapted to human dwellings and their immediate surroundings [47]. The abundance of adult and immature *Ae. aegypti* was higher in urban study sites and mostly in the wet season (May–October). The higher abundance of *Ae. aegypti* in urban settings could be explained by differences in container characteristics and domestic water management [17,48]. Consistent with earlier studies, our study also found that there were more potential breeding containers and containers positive for *Aedes* vectors in urban areas during the wet monsoon season [29]. The predominant breeding sites in our urban sites were also a high number of containers (e.g., tires and discarded containers), while in rural sites, *Ae. aegypti* displayed behavioral plasticity in that the females lay eggs in a vast array of containers, including water storage containers and flower pots. Other risk factors might be construction sites in urban areas, where *Ae. aegypti* seems to be well-suited for reproduction, thus increasing the abundance of breeding sites and density at the neighborhood level [49]. Dengue transmission in Thailand is highly seasonal, with the highest incidence occurring during the rainy season [50]. This may account for the high proportion of houses with water-storage containers found positive for immature *Aedes* mosquitos. Different types of wet containers produced variable numbers of immature *Aedes* throughout the study. Round-shaped and medium-size containers were observed to produce the highest number of *Aedes* larvae and pupae. Water storage jars and tanks are the most commonly used containers in Thailand [17]. Participants use plastic drums, plastic buckets, and water tanks to store water from supplied piped water. Houses that used adult and larval control methods were found to be less infested with female adult and immature *Ae. aegypti* than houses that did not. Outdoor containers showed a higher contribution to dengue vector breeding than indoor containers. Significant numbers of outdoor containers have been reported to be positive with immature *Aedes* in previous studies conducted in Thailand and other countries [17,51,52,53,54]. As people become more aware of the potential oviposition sites of *Aedes* mosquitoes, they usually check and clean the indoor containers located inside their households. Consequently, vectors tend to shift to outdoor containers. This dichotomous treatment behavior was clearly observed in the present study. Such adaptive behavior of *Aedes* mosquitoes poses a severe challenge to vector-control efforts [55,56].

In the present study, most of the sampled sites had high entomological index values indicative of the risk of dengue outbreaks [57,58]. The entomological indices were observed to be higher in the urban areas relative to rural sites, indicating that urban areas are potentially more exposed to dengue risk. There may be a greater risk of dengue infection in urban areas than in rural areas because of the higher population density. Indeed, despite people living in urbanized areas people having better jobs and socio-economic conditions, a greater risk of dengue infections has been reported [59]. In contrast, other studies have found a higher risk of DENV transmission in poorer settings [60,61]. Such associations may depend on the proximity of individuals to risk factors regardless of their level of wealth. The level of education matters, as it is associated with a greater understanding of the principles of hygiene in water and food storage. Traditionally, entomological indices such as the HI, BI, CI, and PI are the chief surveillance tools of many vector-control programs in dengue-endemic countries worldwide [62]. These indices not only measure the success of vector-control strategies but also help to understand the vector ecology. However, the quantifiable association between vector indices and risk of DENV transmission has been questioned in several studies [36,62]. 

In this study, several suites of socio-ecological factors were associated with mosquito abundance. These findings could be readily interpreted and used to inform the design and implementation of targeted vector-control campaigns that reflect local social-ecological contexts. This study found that poor education was related to higher dengue vector infestation. Previous studies in Thailand found an association between risk of dengue and poor education level [16]. Poor climate change knowledge, practices, and attitudes about dengue were associated with higher abundances of *Ae. aegypti*. These findings highlight the importance of educating target populations who have poor education level and KAP on climate change and dengue. This study found a higher ratio of television and internet users among study participants who used such sources of information for climate and dengue. Social media, such as Facebook, Line, and Instagram, were especially used. To increase public awareness regarding the use of preventative measures, we advocate the use of social media and television to disseminate helpful information about adaptation and mitigation measures for climate change, as well as for monitoring and preventing dengue [33,34]. This could be a part of the government vector-control strategy that may benefit from a shift from reactive to proactive vector control. To improve the understanding of community needs and comprehension about dengue and climate, KAP should comprise a key surveillance component of any vector-control program. This allows for the development of culturally appropriate information, trust with community members, and improvements in vector-control activities through active community engagement [63]. 

A greater HCI was also indicative of a greater abundance of *Ae. aegypti* mosquitoes. Household crowding likely reflects the greater risk of exposure to infectious mosquito bites. This finding was consistent with those of other studies, both in Thailand [16] and elsewhere [64]. Our study also suggests that the HCI may be associated with higher densities of population, houses, and water storage containers; distances between houses; average tree height; and average percentage of vegetation cover for each house in urban sites compared to rural sites [48,65]. The PCI was significantly associated with both adult female and immature *Ae. aegypti*. Similar positive associations in Mexico and Brazil between the PCI and immature *Aedes* clearly showed the usefulness of the method [66,67]. Positive correlations between the PCI and house positivity for larvae, pupae, and adult *A. aegypti* led authors to advocate the Brazilian Dengue Control Program for the use of the PCI to schedule the vector-control teams′ visits at different frequencies based on PCI scores [42]. However, evidence about the accuracy of the PCI has been mixed [41]. Household construction may play a role in vector abundance and DENV transmission risk; a previous study found greater risk for contracting dengue among people living in two-floor houses in northeastern Thailand [47]. Interestingly, no such association was found in our study (Table 4), except for significant associations of windows screened with netting and home wall types (wood/cement/bricks) with the abundance of adult female and immature *Ae. aegypti,* respectively.

Global warming has had various direct and indirect effects on human health and infectious diseases. Vector-borne diseases, such as dengue, are forecasted to be most affected by the expansion of areas with vector mosquitoes and an increase in the number and feeding activity of infected mosquitoes [68]. Specifically, climate change has been suggested as one potential contributor to the relative increases in vectorial capacity for dengue vectors and dengue transmission [69]. Studies of the association between climatic variables and dengue vectors are complex, not least because the effects of climate and other environmental changes are location-specific, which can alter the geographic distribution of disease vectors and vector-borne diseases [70,71,72]. In this study, we demonstrated that a generalized linear model with lagged temperature and relative humidity as covariates were key predictors for the abundance of dengue vectors. Our study also suggests that temperature is higher in urban areas and one of the key predictors for the abundance of dengue vectors compared with rural sites. This result is valuable for vector surveillance in dengue-endemic areas with similar climates. There have been several studies on climate in Thailand and other countries that have shown that temperature is one key factor for the distribution of mosquitoes and the transmission of DENV [25,73,74,75]. Temperature modulates DENV epidemic growth rates through its effects on reproduction numbers and generation intervals [76]. The generation interval is highly sensitive to temperature, decreasing two-fold between 25 and 35 °C. Dengue epidemics may accelerate as temperatures increase, not only because of more infections per generation but also because of faster generations [76]. Temperature affects not only the survival rate of mosquitoes but also the lifecycle of the vector, including oviposition, hatching, pupation, and emergence processes [77,78,79]. As temperatures rise, the extrinsic incubation period declines, biting frequency increases, and the average life span of mosquitoes increases [80,81]. According to the Intergovernmental Panel on Climate Change (IPCC), the minimum temperature required for DENV transmission is 11.9 °C and the minimum temperature for the biological activity of *Aedes* mosquitoes is 6–10 °C [82]. It is predicted that at mean temperatures <18 °C, DENV transmission increases as the diurnal temperature range (DTR) increases, whereas at mean temperatures of >18 °C, larger DTR reduces DENV transmission [83]. Previous studies have shown that relative humidity is also a contributing factor for vector abundance and DENV [84,85]. Temperature defines a viable range for transmission; humidity amplifies the potential within that range [25]. Transmission-potential is regulated by temperature-humidity coupling, enabling epidemics in a limited area of weather-space [86]. However, a high relative humidity might be associated with strong rainfall events [87]. The potential impact of changing rainfall patterns on *Aedes* mosquitoes is more difficult to predict since *Aedes* larvae develop in a wide range of water-holding containers, many of which are primarily filled by humans rather than by natural precipitation [88,89]. Any change in climate resulting in the range of meteorological variables conducive for vector breeding is expected to trigger an increase in vector populations and thereby affect DENV transmission. However, the effect of climatic factors on DENV transmission and vector distribution is not consistent throughout the world [84,90,91,92]. Ecological and human factors are essential in driving vector-borne diseases [93].

This study was limited by data that were only collected in household environments and surroundings to assess *Ae. aegypti* habitats and that excluded other habitats in non-household environments. Immature mosquitoes were not collected from all possible breeding sites and might have been under-sampled. DENV detection in mosquitoes was not done; if available, this may have provided additional valuable information about the potential risk for dengue transmission. Despite this limitation and because no such entomology, household, KAP, and climate data were previously available in our study areas of northeastern Thailand, this study provides baseline information on the distribution of dengue vectors, overall KAP of people, and associated risk factors, all of which are essential for planning successful control operations. We expect future research endeavors to attempt to fill in the gaps in our knowledge on the complex dynamics of dengue and its vector from an entomological perspective. 

## 5. Conclusions

The results indicate that low education level, poor socio-economic status, crowded households, poor premise conditions, surrounding house density, unscreened windows, high numbers of wet containers, lack of mosquito control, and prior DENV infections were associated with higher *Ae. aegypti* abundance in the study sites. We also found a strong association between mosquito abundance and household participant′s adaptive capacity to climate change and their practices related to dengue. The current study also shows that maximum and mean temperatures with a lag of four weeks are important meteorological variables that affect *Ae. aegypti* abundance. Understanding the KAP in communities regarding climate change and dengue is essential for improving vector-control and dengue-prevention strategies. In the absence of specific treatment and the incomplete protection provided by the currently available vaccine, these findings may contribute to the development of a reliable early warning system about the potential spread of vector-borne diseases, increasing the awareness of the general public and tourists, as well as promoting community- and individual-level preventive and control measures regarding dengue and its vector in northeastern Thailand and other dengue-endemic countries. The dataset awaits further analysis for the predictive modeling of mosquito abundance and disease risk based on socio-economic, landscape, and temporal patterns, as well as for the development of dengue early warning systems to guide vector-control operations. 

## Figures and Tables

**Figure 1 ijerph-18-05971-f001:**
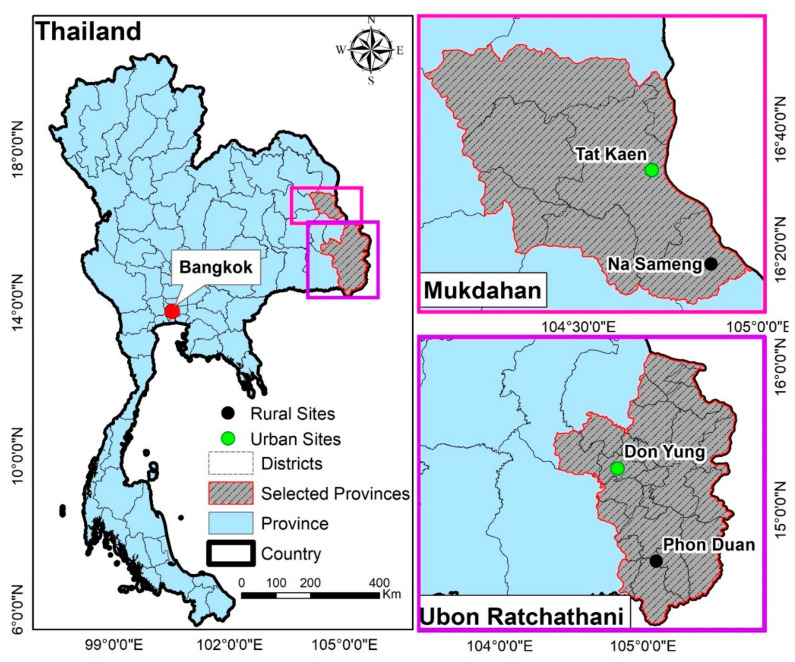
Locations of the four data collection sites in northeastern Thailand. Data were collected from 128 households (32 households per site) in two urban and two rural study sites.

**Figure 2 ijerph-18-05971-f002:**
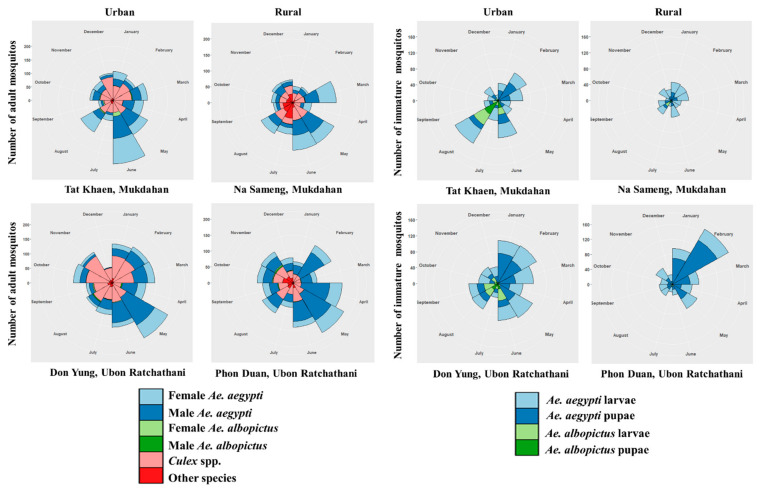
Monthly distribution of mosquitoes caught in a total of 128 households in two urban and two rural study sites in northeastern Thailand during January–December 2019.

**Figure 3 ijerph-18-05971-f003:**
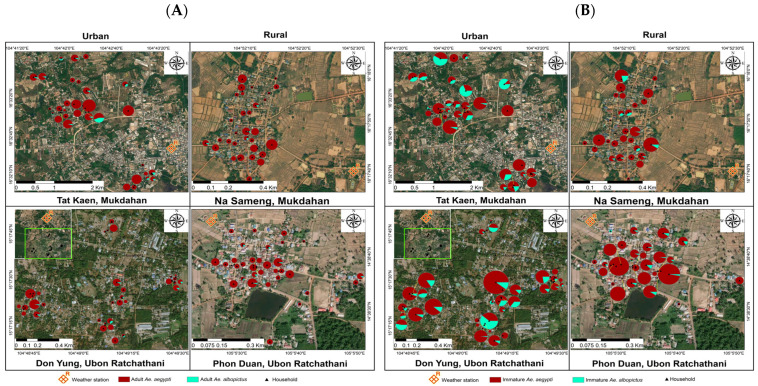
Household distribution of *Aedes* mosquitoes caught in 128 households in two urban and two rural study sites in northeastern Thailand during January–December 2019. The green box represents the inset map of study sites: (**A**) adult mosquitoes; (**B**) immature mosquitoes.

**Figure 4 ijerph-18-05971-f004:**
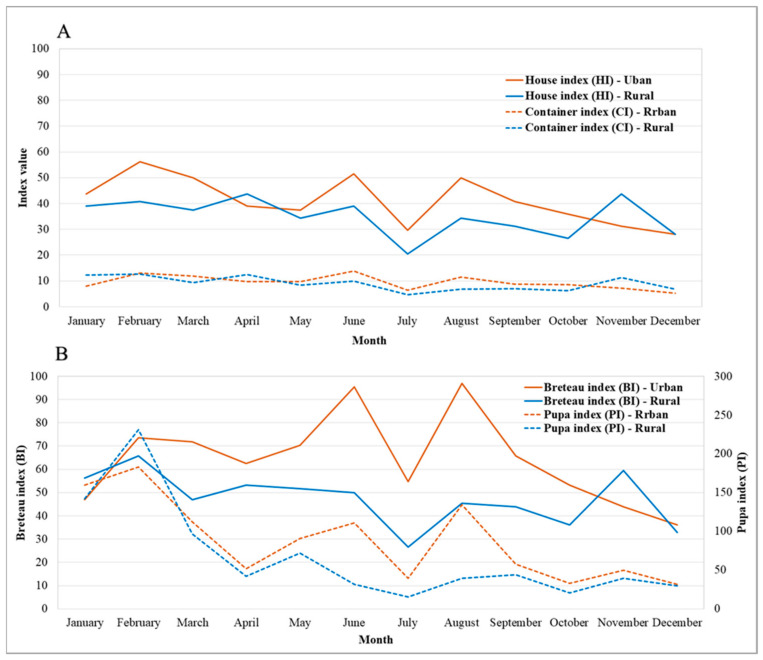
Entomological indices (HI, CI, BI, and PI) in urban and rural study sites in northeastern Thailand during January–December 2019.

**Figure 5 ijerph-18-05971-f005:**
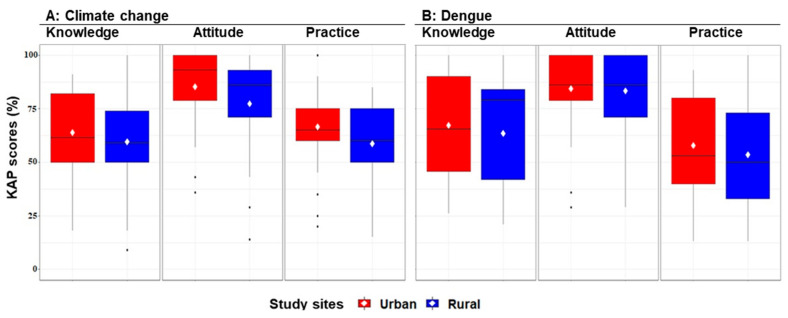
Boxplots showing the percentage and mean scores of knowledge, attitudes, and practices (KAP) in a total of 128 households in urban and rural study sites in northeastern Thailand during February–April 2019. Maximum scores are 100 for each KAP component.

**Figure 6 ijerph-18-05971-f006:**
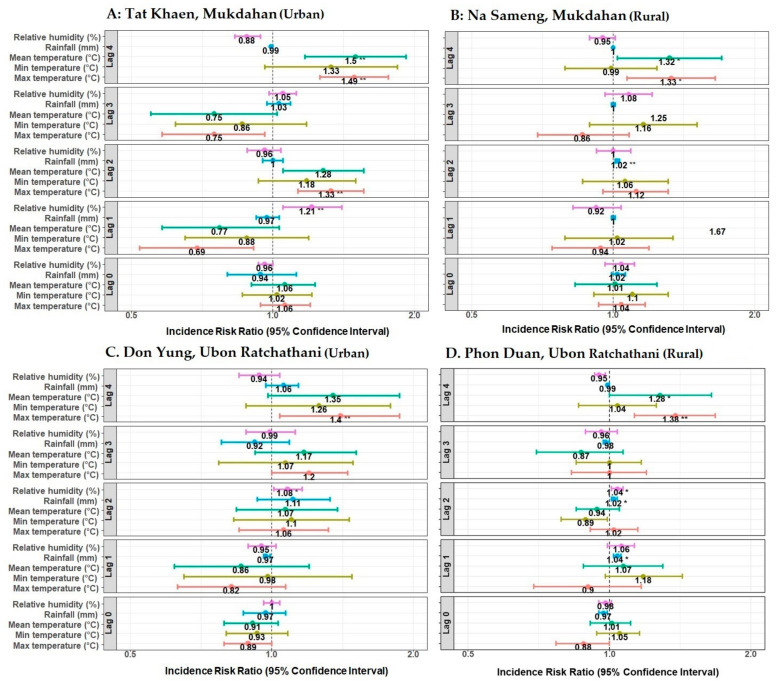
Effect of climate variables on the abundance of adult female *Ae. aegypti* mosquitoes in northeastern Thailand during January–December 2019. Incidence risk ratios were computed by negative binomial generalized linear models. Each panel shows the lag effects (0–4 weeks) of maximum, minimum, and mean temperature (°C); total rainfall (mm); and relative humidity (%). * *p*-value ≤ 0.05, ** *p*-value ≤ 0.01. Each panel (**A**–**D**) represents selected urban and rural study sites.

**Figure 7 ijerph-18-05971-f007:**
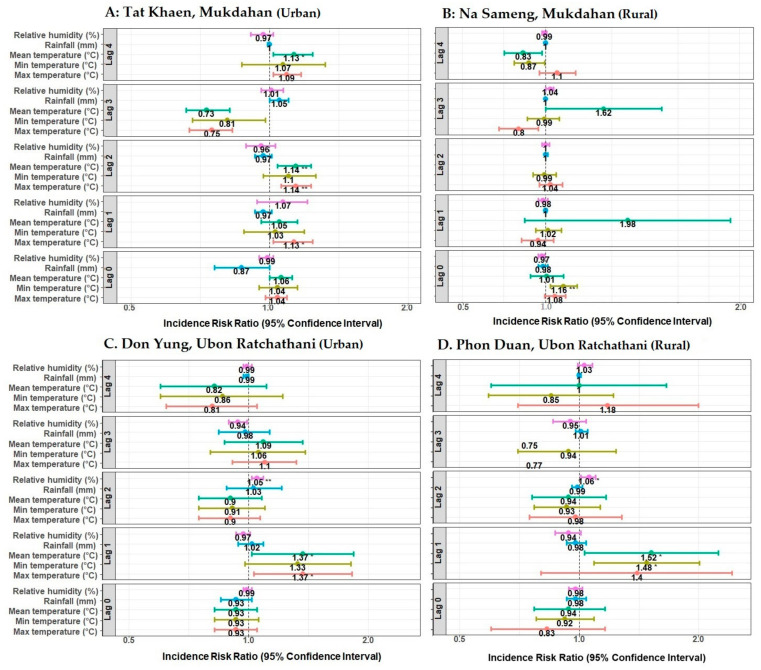
Effect of climate variables on the abundance of immature *Ae. aegypti* mosquitoes in northeastern Thailand during January–December 2019. Incidence risk ratios were computed by negative binomial generalized linear models. Each panel shows the lag effects (0–4 weeks) of maximum, minimum, and mean temperature (°C); total rainfall (mm); and relative humidity (%). * *p*-value ≤ 0.05, ** *p*-value ≤ 0.01. Each panel (**A**–**D**) represents selected urban and rural study sites.

**Table 1 ijerph-18-05971-t001:** Variables used in the premise condition index (PCI), household crowding index (HCI,) and socio-economic status (SES).

Index	Variables	Description	Classification Score
Premise condition index (PCI)	House condition	Good (well-maintained, e.g., newly painted or new house)	1
		Intermediate (moderately well-maintained house)	2
		Bad (not well-maintained house, e.g., paint peeling, broken items visible, and dilapidated old house)	3
	Yard condition	Good (tidy yard)	1
		Intermediate (moderately tidy yard)	2
		Bad (untidy yard)	3
	Shade condition	Not shaded (very little or no shade)	1
		Intermediate (some shade: >25% but <50%)	2
		Shady (plenty of shade: >50%)	3
	Water supply and storage	Piped water	1
		Ground water/well water supply	2
		Rainwater and/or open water source: river/stream/lake/mountain water/river water	3
Household crowding index (HCI)	Co-residents	Monthly number of co-residents per household	-
	Number of rooms	Number of rooms per household	-
Socio-economic status (SES)	House roof material	Ceramic/Wood/Metal	-
	House walls material	Plastered/cement/bricks/wood	-
	Ownership of durable assets	television/VCD/refrigerator/washing machine/mobile/smartphone/computer/oven/microwave/airconditioner/car/pickup/motorcycle	-
	Ownership of toilet facility	Yes/no	-
	Toilet/bathroom floor material	Tiles/cement/earth	-
	Ownership of flush toilet/squat toilet	Yes/no	-

**Table 2 ijerph-18-05971-t002:** Number of mosquitoes caught in 128 households in two urban and two rural study sites during monthly collections in northeastern Thailand during January–December 2019.

Species/Stage	Total Number (%)		Monthly Range, n		Monthly Mean ± SD	
	Urban	Rural	Urban	Rural	Urban	Rural
Adult *Ae. aegypti*						
Female	551 (18.8)	515 (21.8)	6–110	23–81	45.9 ± 28.2	42.9 ± 19.5
Male	823 (28.2)	769 (32.7)	11–166	34–153	64.0 ± 47.3	68.5 ± 36.4
Adult *Ae. albopictus*						
Female	36 (1.3)	11 (0.4)	0–17	0–3	3.0 ± 4.6	0.9 ± 0.9
Male	16 (0.6)	12 (0.5)	0–5	0–4	1.3 ± 1.6	1.0 ± 1.1
*Culex* spp.	1302 (44.6)	677 (28.8)	63–149	40–83	108.5 ± 26.5	56.4 ± 13.6
Other species	191 (6.5)	370 (15.8)	6–27	6–54	15.9 ± 5.9	30.8 ± 19
Total adult mosquitoes	2919 (100)	2354 (100)	0–166	0–153	39.8 ± 45.8	31.6 ± 27.9
Immature *Ae. aegypti*						
Larvae	647 (42.7)	525 (48.8)	33–75	23–65	53.9 ± 14.4	43.7 ± 11.3
Pupae	543 (35.8)	495 (46.0)	16–112	10–148	16.4 ± 17.1	3.0 ± 3.8
Immature *Ae. Albopictus*						
Larvae	197 (12.9)	37 (3.4)	0–50	0–14	45.2 ± 31.7	41.2 ± 39.3
Pupae	131 (8.6)	19 (1.8)	0–57	0–10	10.9 ± 16.3	1.5 ± 2.7
Total immature mosquitoes	1518 (100)	1076 (100)	0–112	0–148	33.4 ± 32.2	22.4 ± 28.8

SD: standard deviation.

**Table 3 ijerph-18-05971-t003:** Number of immature *Aedes* mosquitoes (%) collected in containers in 128 households in two urban and two rural study sites in northeastern Thailand during January–December 2019.

		Larvae		Pupae		Total
Container Characteristics	Description	Urban	Rural	Urban	Rural	
Shape of container	Square (Cemented tank, flower vase/pots)	27 (7)	66 (24)	7 (6)	21 (26)	121 (14)
	Round (jar, bucket, etc.)	292 (79)	198 (71)	99 (79)	54 (66)	643 (75)
	Other (tree holes, bamboo, ant traps, solid waste, etc.)	51 (14)	15 (5)	19 (15)	7 (8)	92 (11)
Size of container	Small (<50 cm)	161 (44)	97 (35)	56 (45)	31 (38)	345 (40)
	Medium (50–100 cm)	200 (54)	172 (61)	66 (53)	46 (56)	484 (57)
	Large (>150 cm)	9 (2)	10 (4)	3 (2)	5 (6)	27 (3)
Container Cover	Good	12 (3)	3 (1)	2 (2)	2 (2)	19 (2)
	Poorly fitted	34 (9)	15 (5)	14 (11)	7 (9)	70 (8)
	None	324 (88)	261 (94)	109 (87)	73 (89)	767 (90)
Location	Indoor	120 (32)	129 (46)	37 (30)	42 (51)	328 (38)
	Outdoor	250 (68)	150 (54)	88 (70)	40 (49)	528 (62)
In toilet or not	In toilet	108 (29)	165 (59)	35 (28)	42 (51)	350 (41)
	Not in toilet	262 (71)	114 (41)	90 (72)	40 (49)	506 (59)
Larval control types	Abate	20 (5)	12 (4)	5 (4)	4 (5)	41 (5)
	Larval control washed in last week	28 (8)	42 (15)	8 (6)	7 (9)	85 (10)
	No larvae control	322 (87)	225 (81)	112 (90)	71 (86)	730 (85)

**Table 4 ijerph-18-05971-t004:** Incidence rate ratios (IRRs) for the abundance of *Ae. aegypti* per household in relation to socio-demographic and household risk factors using negative binomial generalized linear models. Data were collected from 128 households in northeastern Thailand during January–December 2019.

		Female Adults		Immatures	
Variables	*n* (%)	IRR (95% CI)	*p*-Value	IRR (95% CI)	*p*-Value
Sites types					
Urban	64 (50)	1.55 (1.21–1.98)	0.000	1.47 (1.01–2.17)	0.042
Rural	64 (50)	1		1	
Education level					
<=Primary	89 (69.5)	1.39 (1.07–1.79)	0.011	1.49 (1.02–2.18)	0.032
>Primary	39 (30.5)	1		1	
Socio-economic status					
Poor	36 (28.1)	0.94 (0.72–1.23)	0.693	2.15 (1.39–3.32)	0.001
Intermediate	52 (40.6)	1.03 (0.81–1.3)	0.801	1.63 (1.1–2.43)	0.015
Wealthy	40 (31.3)	1		1	
Household crowding index (HCI)					
3 (Crowded)	31 (24.2)	1.76 (1.27–2.43)	0.001	0.75 (0.46–1.23)	0.263
2 (Medium crowded)	65 (50.8)	1.58 (1.22–2.05)	0.000	0.50 (0.34–0.74)	0.001
1 (Not crowded)	32 (25.0)	1		1	
Premise condition index (PCI)					
9–10 (High)	41 (32.0)	1.97 (1.49–2.61)	0.000	1.54 (1–2.37)	0.043
7–8 (Medium)	46 (36.0)	1.20 (0.91–1.59)	0.179	1.07 (0.72–1.59)	0.730
5–6 (Low)	41 (32.0)	1		1	
House density (houses per km^2^)					
100–200	8 (6.3)	1.29 (0.83–1.99)	0.246	0.67 (0.34–1.34)	0.267
201–500	38 (29.7)	1.37 (1.05–1.8)	0.021	0.84 (0.55–1.29)	0.433
501–1000	47 (36.7)	1.19 (0.92–1.55)	0.167	1.49 (1.02–2.18)	0.038
>1000	35 (27.3)	1		1	
House type					
Single house, one family, two floors	64 (50)	1.009 (0.82–1.22)	0.931	1.27 (0.91–1.77)	0.146
Single house, one family, one floor	64 (50)	1		1	
Roof materials type					
Metal (e.g., corrugated iron)	97 (75.8)	1.14 (0.83–1.58)	0.400	0.89 (0.54–1.46)	0.659
Wood	17 (13.3)	1.03 (0.65–1.61)	0.889	0.57 (0.28–1.16)	0.125
Ceramic	14 (10.9)	1		1	
Wall type					
Wood	12 (9.4)	1.11 (0.79–1.57)	0.530	1.68 (1.01–2.81)	0.045
Cement/bricks	33 (25.8)	0.95 (0.76–1.18)	0.652	1.48 (1.05–2.09)	0.022
Plastered	83 (64.8)	1		1	
Location of bathroom/toilet					
Indoors	91 (71.1)	1.46 (1.14–1.89)	0.003	1.01 (0.68–1.5)	0.925
Outdoors	37 (28.9)	1		1	
Bathroom floor type					
Cement	54 (42.2)	1.15 (0.91–1.47)	0.221	0.73 (0.5–1.07)	0.109
Tiles	74 (57.8)	1		1	
Eaves status					
Closed	79 (61.7)	1.14 (0.92–1.42)	0.201	0.94 (0.67–1.3)	0.715
Opened	49 (38.3)	1		1	
Windows					
Unscreened	107 (83.6)	1.41 (1.04–1.92)	0.025	1.65 (1.05–2.6)	0.029
Screened	21 (16.4)	1		1	
Number of wet container					
>50	109 (85.2)	1.33 (1.01–1.75)	0.037	1.41 (0.92–2.16)	0.112
<50	19 (14.8)	1		1	
Use any kind of larvae control					
No	93 (72.7)	0.93 (0.75–1.14)	0.503	1.20 (0.84–1.7)	0.300
Yes	35 (27.3)	1		1	
Use any kind of adult control					
No	72 (56.2)	1.24 (1.01–1.55)	0.045	1.13 (0.79–1.61)	0.496
Yes	56 (43.8)	1		1	
Self-reported dengue infections					
Yes	12 (9.4)	1.68 (1.22–2.32)	0.001	0.79 (0.46–1.35)	0.398
No	116 (90.6)	1		1	
Climate change knowledge					
Poor	97 (75.8)	0.87 (0.61–1.24)	0.451	1.97 (1.19–3.25)	0.008
Good	31 (24.2)	1		1	
Climate change attitude					
Poor	49 (38.3)	0.97 (0.79–1.19)	0.801	0.71 (0.52–0.99)	0.043
Good	79 (61.7)	1		1	
Climate change practice					
Poor	105 (82.0)	1.52 (1.07–2.16)	0.017	1.84 (1.13–2.99)	0.014
Good	23 (18.0)	1		1	
Dengue knowledge					
Poor	73 (57.0)	0.91 (0.7–1.17)	0.491	1.18 (0.8–1.75)	0.391
Good	55 (43.0)	1		1	
Dengue attitude					
Poor	40 (31.3)	1.24 (1.01–1.53)	0.035	0.75 (0.54–1.04)	0.092
Good	88 (68.8)	1		1	
Dengue practice					
Poor	94 (73.4)	1.43 (1.03–1.99)	0.029	1.93 (1.17–3.18)	0.009
Good	34 (26.6)	1		1	
Model fit					
Omnibus test	131.2		0.000	957.2	0.000
AIC	719.2			935.3	
BIC	810.5			1026.6	

CI: confidence interval; SD: standard deviation; AIC: Akaike′s information criterion; BIC: Bayesian information criterion.

**Table 5 ijerph-18-05971-t005:** Monthly climate variables in urban and rural sites in northeastern Thailand during January–December 2019.

Meteorological Variables	Range (n)		Mean ± SD	
	Urban	Rural	Urban	Rural
Mean temperature (°C)	23.2–32.6	22.7–31.0	28.0 ± 2.3	27.2 ± 2.2
Minimum temperature (°C)	13.2–26.7	9.6–23.7	21.6 ± 3.7	19.1 ± 3.9
Maximum temperature (°C)	31.0–41.8	33.0–41.8	35.5 ± 3.0	37.0 ± 2.6
Relative humidity (%)	58.0–88.1	60.0–88.7	73.5 ± 8.1	76.3 ± 8.2
Total rainfall (mm)	0–803.3	0–1229.4	102.9 ± 173.3	166.5 ± 254.7

SD: standard deviation.

## Data Availability

The data underlying the results presented in the study are available from the Norwegian Center for Research Data (NSD). Access to the data sets must be requested from NSD using a Data Access form at this link: https://nsd.no/nsd/english/order.html and referring to project ID NSD0000 (accessed on 22 April 2021).

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
