# Peer review of "Ecological, Social, and Other Environmental Determinants of Dengue Vector Abundance in Urban and Rural Areas of Northeastern Thailand"

_ijerph, 2021, doi:10.3390/ijerph18115971_

Round 1

Reviewer 1 Report

This is an ambitious and interesting manuscript describing a one-year entomological study conducted in four villages of northeastern Thailand.  Overall, it was found that participants' KAP about climate change and dengue were low in both urban and rural areas. There is mention of a previous KAP study but in this paper, although supplementary information is provided, the KAP study isn't well explained and the results not fully presented. It would be good to summarize current risk mitigation practices.

Most of the figures are hard to read and could be improved. The study is complex and includes a lot of interesting data and so the Tables and Figures could be more self explanatory.

Although it was found that urban sites had higher numbers of adult Ae. aegypti mosquitoes than those in a rural setting and that the fitted GLM indicated that a higher abundance of adult female Ae. aegypti was associated with a wide range of factors, these factors are not clearly explained. Was the sampling effort equal in the urban and rural settings ? 

It is reasonably concluded that the low KAP of respondents regarding climate change and dengue indicates the need for more education and outreach on vector-borne disease prevention in this region. As this is an important outcome, the data with regard to KAP in the study participants could be better presented.

There are a few minor grammatical errors which should be addressed.

Reviewer 2 Report

Interesting, well designed study.

As the authors rightly state, the incidence of dengue has dramatically spread and increased globally in the past 40 -50 years. Approximately half of the world’s population is at risk of contracting the disease, with an estimated 390 million infections occurring annually in 128 countries.

I would like to pay attention to weak points, in my opinion:

  1. Results table 4 - standardize the notation p
  2. Discussion:

-A greater risk of dengue infection in urban areas than in rural areas because of the higher population density. Although they are treated as richer. The level of education matters - which is associated with a greater understanding of the principles of hygiene in water and food storage. Data on the relationship with often, as previously published.

- A very important observation that there is a relationship between global warming and the incidence of dengue fever - in my opinion, it requires a broader comment in the discussion.

-The strengths of the study is a large, representative sample from a survey in Thailand and adequate statistical methodology. Please describe in the discussion: limitation of the study

Round 2

Reviewer 1 Report

The manuscript is much improved. Figure 1 is a bit untidy and the writing in parts of figures 3 and 5 is hard to read. A grammar and spell check is needed to remove duplication throughout the text especially where edits have been made. Line 332, man could be replaced by 'human'

Author Response

The manuscript is much improved.

Point 1: Figure 1 is a bit untidy and the writing in parts of figures 3 and 5 is hard to read.

Response 1: Changes were made accordingly in figures 1, 3, and 5 for better visualization and understanding.

Point2: A grammar and spell check is needed to remove duplication throughout the text especially where edits have been made.

Response 2: Grammar and spell were checked throughout the paper.

Point 3: Line 332, man could be replaced by 'human'

Response 3: In line 335, the correction was made according to the suggestion.

Reviewer 2 Report

The authors corrected the text in line with my comments. I accept the manuscript for publication.

Author Response

The authors corrected the text in line with my comments. I accept the manuscript for publication.

Thanks.